# A qualitative study of graduate student emotional and cognitive processing of unexpected (chance) events

Hope Ferguson®*, Elisabeth E. Schussler

Department of Ecology and Evolutionary Biology, The University of Tennessee, Knoxville, Tennessee, United States of America

* hjohns76@vols.utk.edu

## Abstract

Research on graduate education often focuses on the expected challenges of graduate programs, such as proposal defenses, comprehensive exams, and mentoring relationships. However, less is known about how unexpected (chance) events may shape graduate student program experiences, including their impacts on career intentions. Prior research suggests that unexpected events introduce uncertainty and opportunities, however, the emotional and cognitive processes underlying these perceptions are unknown. This study explored how graduate students emotionally and cognitively processed a self-reported highly impactful chance event during their program. Specifically, this study asked 1) How did life science graduate students initially appraise and reappraise chance events? and 2) How did the narratives of chance event appraisals differ among participants who changed their career intentions versus those who did not? Using Appraisal Theory, we conducted thematic narrative analysis for the interviews with ten current life science graduate students. Five students had changed their career intentions since starting graduate school and five had not. These analyses revealed that themes of *Initial Stressors, Initial Emotional Toll,* and *Growth* were shared among participants, highlighting these as common processes related to unexpected events. However, the *Help-Seeking, Self-Reliance, Realizations,* and *Resilience* themes differed among those who had changed their career intentions or not. Those who had changed their career intentions often coped by using external support and prioritizing their mental well-being. Those who had not changed their career intentions coped via self-reliance, focusing on self-managing challenges and viewing chance events as opportunities to build new skill sets. This study demonstrated the utility of applying Appraisal Theory to graduate study, showcasing how appraisal and reappraisals of unexpected events may be related to career decision making. Implications include the need for effective coping strategies to navigate unexpected challenges and faculty and departmental support for career development.

**Data availability statement:** De-identified data are available in the Supporting Information (S4 Appendix). This includes participant quotes and narrative summaries used for the analysis.

**Funding:** The author(s) received no specific funding for this work.

**Competing interests:** The authors have declared that no competing interests exist.

## Introduction

A large body of research on challenges faced by science graduate students in their degree programs has focused on outcomes such as students leaving their programs [1], however, a more common outcome for many students is for them to shift their career intentions while in their programs [2–7]. Research has suggested a connection between these career intention shifts and challenging aspects of the graduate student experience, such as program requirements (i.e., proposal writing, preliminary exams, resource management, or dissertation submission deadlines), mentoring or supervisory relationships, low self-efficacy and motivation, stereotype threat and stigmatized identities, or lack of family-friendly policies [8–10]. These experiences are frequently framed as expected challenges that graduate students must overcome to proceed toward their degree and their anticipated future careers. However, sometimes unexpected things happen [2], and research has only begun to examine how these unanticipated occurrences might impact graduate students, including how they interpret and process these experiences in relation to career intentions.

Unplanned or unexpected events, 'chance events,' can significantly impact the experiences and career intentions of graduate students [2,11,12]. What constitutes a chance event can differ from person to person, depending on what they consider to be expected, or within their locus of control, preparedness, and impact [13,14]. However, researchers have identified categories of frequently perceived chance events such as unexpected professional or personal encounters, historical events (i.e., the COVID-19 pandemic), unforeseen career obstacles such as illness or discrimination, or financial problems [15]. What qualifies as a chance event, though, is not only shaped by the type of event but also how an individual interprets the event within their personal or academic context. A student who anticipated financial struggles, for example, would not call a reduction in funding a chance event, while another student may be surprised because other students never talked about this happening to them. This difference in expectations may then impact how they react to and manage the situation.

Until recently, the impact of chance events on graduate students career intentions had not been thoroughly investigated. Ferguson and Schussler [2] conducted a survey of 39 life science graduate students at one institution and found participants self-reported, on average, three chance events that influenced their career intentions while in their current program. The types of chance events include being in the right place at the right time, unexpected historical events, or unexpected personal or professional encounters. When these participants were asked to rate these events as having a positive, negative, both positive and negative, or neutral impact on their career intentions, almost half indicated they were positive, and none perceived them as solely negative. This was despite disruptions–like the COVID-19 pandemic–often being seen as having negative implications on graduate student mental health and research progress [16]. Open-ended responses on the survey suggested that although the graduate students saw the chance events as presenting challenges, they also thought they offered insights, provided opportunities, and inspired reflection

about their careers. Thus, despite experiencing something unexpected and surprising, the students seemed to have reframed the events as positive impacts and leveraged them to inform their career intentions. However, the short survey responses did not allow insight into how these students emotionally and cognitively transformed these experiences into more positive impacts. Understanding these processes is crucial because emotional and cognitive responses have been theoretically linked to how individuals reassess their goals, adapt, and make career changes [17].

Emotional and cognitive processing are co-occurring reactions to external stimuli. This processing is shaped by the individual's background and prior experiences as well as their current context—including program expectations, institutional norms, and the potential shared experiences of other students [18]. Emotional processing refers to how an individual experiences, expresses, and thinks about their feelings in response to an event [18,19]. These feelings are associated with cognitive (e.g., feeling angry or anxious) and physical (e.g., heart racing) impacts. Importantly, emotional processing is related to behavioral outcomes through reflection on how an event may impact what an individual wants to achieve [17]. Cognitive processing involves reasoning and evaluation. In cognitive processing, individuals assess the emotional significance of their experiences and decide how to respond to them [17], which involves selecting coping strategies.

Coping strategies can be adaptive (e.g., support seeking or problem solving) or maladaptive (e.g., rumination or social withdrawal) and are influenced by students' current environment (e.g., the support offered or structure of coursework), mindsets, and previous experiences [20]. The strategies they select can either support (adaptive) or hinder (maladaptive) their well-being and progress. Two graduate students may experience a similar failure conducting an experiment, leaving both feeling self-doubt and negative emotions. However, as they engage in cognitive processing, one may assess the situation and choose an internal coping strategy, such as problem-solving or self-talk, while the other may choose an external coping strategy, such as seeking help from their advisor. While research can help identify which coping strategies were selected in relation to contextual factors—and how effective the strategies may be—the reasoning of these choices is difficult to explain without understanding how they make meaning of the event [18].

This study interviewed current life science graduate students who had indicated on a survey that they had experienced a chance event that impacted their career intentions. Our goal was to understand how they made meaning from the experience with the chance event, specifically focused on the emotional and cognitive processing related to their career intentions.

## Theoretical framework

This study used Appraisal Theory, developed by Lazarus and Folkman [18], to study how graduate students perceived and responded to chance events. Appraisal theory is a foundational framework that examines how an individual emotionally and cognitively processes pivotal events. This theory acknowledges that responses to events—especially stressful ones—are shaped by static traits (e.g., personality characteristics) but also personal contexts (e.g., past experiences, time in program, health, institutional culture). For example, prior undergraduate experiences or different levels of graduate mentor support may influence how graduate students respond to chance events. However, Lazarus and Folkman [18] explain that focusing on these contextual features cannot fully explain how they process or cope with these events in real time. Coping is a dynamic process which requires individuals to actively make sense of events as they evolve. Therefore, Appraisal Theory provides a structure to analyze how individuals evaluate and respond to chance events based on emotional and cognitive processing, emphasizing interpretations that guide behavior (such as career intentions) and responses to events in the future.

Appraisals occur through a two-part appraisal process (Fig 1). In primary appraisals, emotional processing occurs as individuals assess the event's significance in relation to their goals and values. In secondary appraisal, individuals use cognitive processing to evaluate their available resources (personal strengths, skills, and social support) and their perceived coping ability. This involves reflecting on previous experiences, assessing their self-efficacy, or considering their confidence levels [21]. Primary and secondary appraisal can happen within similar time periods, but this is not the only

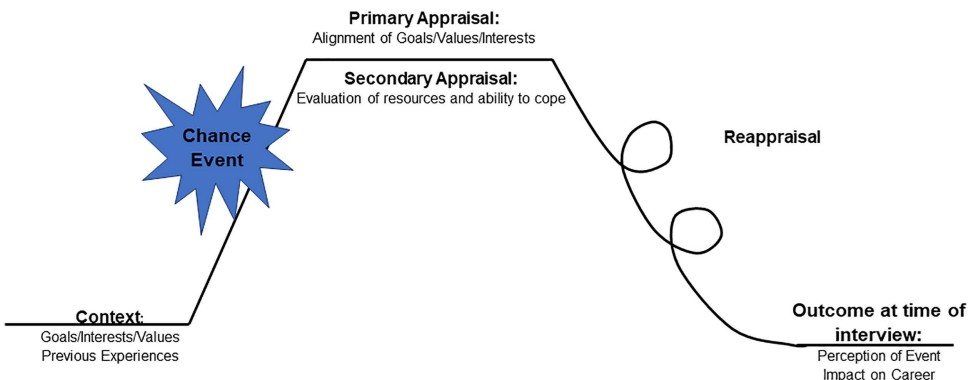

**Fig 1. A schematic representation of Appraisal Theory.** Individuals have their own goals, interests, and values that are guided by their previous experiences. When a chance event occurs, they experience a primary appraisal, where individuals assess the event's relevance and impact, followed by secondary appraisal, where they evaluate their ability to cope. Reappraisals occur when individuals reassess the event based on new information or changing circumstances, ultimately leading to outcomes.

opportunity for an individual to react to and reflect on the event. Individuals also reassess the event's impact, known as re-appraisals [22]. These ongoing evaluations can retrospectively change how individuals perceive and feel about the event, shifting what they think about its impact over time [22].

When paired with interviews, Appraisal theory is a powerful tool to investigate individual processing of events. One study on K-12 teachers, for example, used interviews to create narratives of how they appraised their experiences with emotional exhaustion, depersonalization, personal accomplishment, well-being, and resilience. They found that reactions were guided by goal congruence, coping potential, and the responsibility they felt toward the experience [23]. This study highlighted the complexity of the appraisal process and its value in examining how people evaluate, process, and respond to events; however, it has rarely been applied to graduate students. One study conducted by Yu-Whattam [24] used Appraisal Theory to examine Chinese doctoral students' experiences writing their thesis in an English-speaking country. This study focused on the emotional experiences of the participants during the writing process, examining what emotions they felt, what triggered these emotions, and the coping strategies they used to manage them. Yu-Whattam [24] found that students tended to self-regulate or seek academic support to cope with their emotions. This study demonstrated the utility of using Appraisal Theory to study graduate student experiences; however, it only addressed initial appraisals and not the process of reappraisal. Our study used interviews to explore how graduate students perceived, processed, and responded to chance events in relation to their career intentions.

## Rationale and research questions

Research on events that may contribute to graduate student career intention changes has largely ignored the impact of chance events. Our previous study found that graduate students self-reported experiencing chance events that impacted their career intentions [2], and open-ended responses suggested that these events were viewed as positive agents of change. Yet the types of cognitive and emotional processing that students underwent in response to these events were unknown. Also unknown was whether cognitive and emotional processing might differ for students who shifted their career intentions since starting graduate school versus those who did not. To investigate this, we interviewed life science graduate students from our survey study who indicated that a chance event had significantly impacted their career. Half of the participants had changed their career intentions since starting graduate school, and half had not. Given variability in individual student expectations and graduate school norms, we allowed graduate students to self-define what a chance event was to them because the intent was to explore their reaction to an unexpected event in graduate school. Our research

was guided by Appraisal Theory, which we used in conjunction with thematic and narrative analyses to answer the following questions:

1. How did life science graduates initially appraise and reappraise chance events?

2. How did the narratives of chance event appraisals differ among participants who changed their career intentions versus those who did not?

In answering these questions, we can better understand how the graduate students navigated and adapted to the unexpected and how this may be related to reconsidering their career intentions. Understanding this can inform the development of more effective career guidance and support systems for graduate students.

## Methods

### Recruitment and participants

This research was part of a larger study on life science graduate students and chance events that began with a survey in the Fall of 2021 and invited respondents to participate in interviews in 2022. The University of Tennessee Institutional Review Board (IRB) approved the study (IRB-21–06521), which included a combined consent process for the survey and follow-up interviews. The consent was presented at the beginning of the survey, requiring participants to click yes or no; interview participants then verbally re-confirmed their consent before the interviews. This allowed the data from the surveys to be used as part of the interviews.

Thirty-nine students completed the survey, which asked them to self-report the types of chance events they had experienced in graduate school and their impacts on their career intentions [2]. Participants were given a definition of what was meant by 'chance events' (i.e., an unplanned or unpredictable event) and then given several chance event types from previous research to guide their responses. The participants were current PhD or master's students in a life science program (i.e., Ecology and Evolutionary Biology, Microbiology, Agriculture, or Biochemistry & Cellular and Molecular Biology) at one research institution. At this research institution, both PhD and master's programs are similarly structured, but differ in the number of research chapters. Therefore, both PhD and master's students were included to capture a broader range of potential experiences in graduate school. This population was selected for their accessibility and the author's familiarity with these programs which provided valuable perspective during data analysis. Demographically, graduate students at this institution are similar in age range and mostly white. Of those who took the survey, 29 participants indicated they would be interested in follow-up interviews for this study.

From this list of 29, we selected participants who indicated in the survey that they had experienced at least one chance event that highly impacted their career (Table 1). We then identified individuals from this group who had indicated a change in their career intentions since starting graduate school and who had not changed their career intentions since starting graduate school. We sent interview invitations in late spring/ early summer 2022 to each pool until we reached five participants from each group. Once these participants reaffirmed their willingness to be interviewed, they were provided with possible dates and times for the interview. All interviews were conducted between July and September 2022 using Zoom teleconferencing software. The participants who completed the interviews were provided $10 to compensate them for their time. Authors removed identifying information from the transcripts during data analysis, assigning pseudonyms to each participant.

Participants were all between the ages of 25−34, were mostly women (N = 7), white (N = 8), seeking a PhD (N = 8), and in the 4th year of their programs (N = 5) (Table 1). Six participants were from the same life science department, while the remaining four were from different life science departments. All interviews were conducted in the context of the COVID-19 pandemic, which shaped the academic environment and ultimately the way participants may have experienced and processed chance events. The highly impactful chance events that participants experienced were examples from four of

**Table 1. Participant demographics, career intention shifts since starting graduate school, and the chance event that had the highest impact on their career intentions. Participant names are pseudonyms.**

| Participant | Gender | Race | Highly Impactful Chance Event | Career Intention Shift |
|---|---|---|---|---|
| **Yellow**<br>*2nd Year PhD* | Man | White | Historical Event<br>• Interaction with the COVID-19 pandemic & question of research importance | No Career Shift<br>• Interest in career as principal investigator or faculty member persisted |
| **Bronze**<br>*1st Year PhD* | Woman | White | Historical Event<br>• COVID-19 pandemic | No Career Shift<br>• Interest in career as a research scientist for a non-profit persisted |
| **Khaki**<br>*4th Year PhD* | Woman | White | Obstacles in Career Path<br>• Advisor moved to new university | No Career Shift<br>• Interest in a research-intensive career in biotechnology or pharmacy persisted |
| **Pearl**<br>*4th Year PhD* | Woman | White | Obstacles in Career Path<br>• Negative feedback from students | No Career Shift<br>• Interest in career as principal investigator persisted |
| **Olive**<br>*3rd Year PhD* | Man | White | Historical Event<br>• COVID-19 pandemic | No Career Shift<br>• Interest in a research-intensive career in government persisted |
| **Beige**<br>*3rd Year Masters* | Woman | White | Unexpected Job Opening<br>• Job posting requirements | Career Shift<br>• Initial interest in bench science switched to policy and education for general public positions |
| **Aqua**<br>*4th Year PhD* | Man | Non-white | Professional/Personal Encounter<br>• Meeting fellowship mentor | Career Shift<br>• Interest in academic career switched to careers in industry |
| **Violet**<br>*2nd Year Masters* | Woman | White | Obstacles in Career Path<br>• Diagnosis of panic disorder | Career Shift<br>• Interest in bench science switched to business of science (i.e., project manager) |
| **Lime**<br>*4th Year PhD* | Man | White | Unexpected Job Opening<br>• Teaching position at high school | Career Shift<br>• Interest in academic career switched to education for K-12 |
| **Coffee**<br>*4th Year PhD* | Woman | Non-white | Historical Event<br>• Negative feedback on progress report due to setbacks of COVID-19 | Career Shift<br>• Interest in careers in academia switched to writing related careers in science |

the chance event types provided in the survey: historical events, career obstacles, unexpected job openings, and personal or professional encounters [2] (Table 1). Four of the specific chance events were related to disruptions surrounding the COVID-19 pandemic (i.e., historical events), which included the unexpected duration or impact of the pandemic, shifts in research projects, and negative feedback on progress. Other specific chance events included unexpected health diagnoses, an advisor moving to a different university, or negative teaching feedback (i.e., obstacles in career path).

## Data collection

The questions for the semi-structured interview protocol were developed and informed by Appraisal Theory, and participants were asked about the chance events they identified in the survey [2]. The questions focused on how participants emotionally and cognitively processed and responded to (appraised and reappraised) the event in relation to their career intentions. The draft interview protocol was tested and revised through four pilot interviews to ensure the questions were clear and elicited the desired information. These pilot interviews were not included as part of the results of this study. The final interview protocol can be found in the S1 Appendix.

The interviews first asked participants to rank the chance events they had mentioned in the survey from highest to lowest impact on their career. The participants were then asked about the three highest impact events in relation to their career intentions. Only the most impactful chance event for each participant was used for this study. For this chance

event, the interviewer asked them to describe what was happening in their program before the event occurred (the proximal context) and then asked them to describe the details of the event. They were asked how they felt when the event happened as well as how they felt about it a few months later and then now. They were also asked about the level of control they felt they had over the event and why it had an impact on their career. Although some contextual information was gathered, such as what was happening in their program before the chance event, the interview was focused on the emotional and cognitive reactions to the events rather than examining the contextual factors that may have shaped these reactions. Furthermore, while the participants described the chance events in detail, the focus of the study was on the way the participants reacted to and processed the unexpected occurrence over time, versus the event itself.

The first author, H.F., conducted all interviews. Each interview lasted between 60–120 minutes. All interviews were audio-recorded and transcribed using Zoom teleconferencing software (zoom.us).

## Positionality

The first author, H.F., was formally trained in qualitative research methods and worked on several prior published studies using these methods. H.F. is a PhD student at the institution where the study was conducted and is a student in the life science department (with a dissertation focused on biology education research), where some of the participants were also students. H.F., therefore, had an insider perspective that influenced her interest in researching the experiences of graduate students and their responses to chance events. While this shared experience provided valuable context for interpreting student narratives, it introduced potential bias. Acknowledging this potential bias, H.F. used open-ended questions in the interviews to allow participants to share their own experiences and worked to ensure their voices were represented throughout the analysis. This approach helped avoid influencing participants' responses, providing space for personal insights and more genuine reflections [25]. We used several strategies to ensure transparency and reflexivity as per best practice [26]. This included the creation of memos during data collection and analysis which, allowed for reflection on how positionality may have shaped how data were interpreted.

## Data analysis

This study used a Thematic Narrative Analysis approach [27,28], which is a combination of narrative analysis and thematic analysis, to investigate the processing of chance events by graduate students. Narrative analysis examines how individuals reflect on and make sense of their life experiences through storytelling [29,30]. This method recognizes that people actively construct their identities by narrating stories about their lives [31,32]. When individuals shared their experiences with chance events, they highlighted certain salient aspects of their experiences—such as achievements, failures, and challenges they had to overcome or personal growth—while leaving other factors out [30]. This selective storytelling is what shapes their personal life stories and serves as their personal explanations for their actions.

We used thematic narrative analysis in a stepwise approach (Fig 2). To address research question one, we used narrative analysis to create individual, primary narratives for each participant. These are referred to as Appraisal Theory Story Arcs – see description below. Individual narratives then underwent thematic narrative analysis to identify recurring themes that reflected how the participants made meaning of the event. To address research question two, we compiled the themes and narratives from each participant into two core composite narratives: one for participants who changed their career intentions and one for participants who did not. This allowed us to compare themes among the two groups. De-identified data used for this analysis are provided in the Supporting Information (S4 Appendix).

**Research question 1: How did life science graduate students initially appraise and reappraise chance events?** The individual narratives that encompassed the graduate students' reactions to their chance event were created from their interviews. Due to the probing of participant ideas, the transcript was not a chronological account of their story; thus, the transcripts needed to be reorganized into narratives before analysis could occur. To create the narratives, we used the Appraisal Theory framework to create a story arc, which have structural components such as exposition, rising

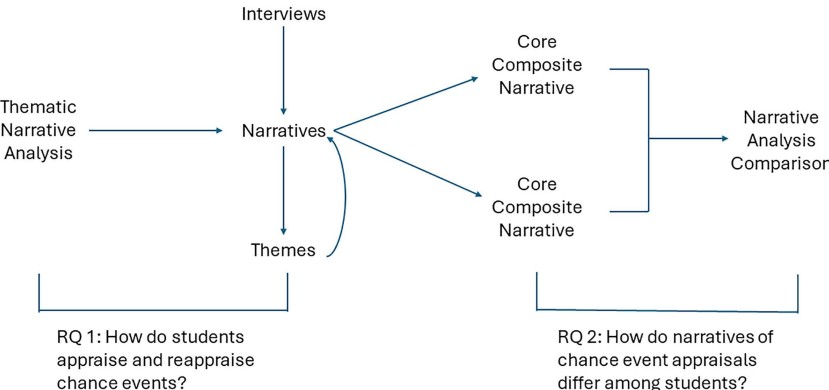

**Fig 2. Sequence of methods.** The sequence of the research methods began with interviews that were analyzed through thematic narrative analysis to generate individual narratives and identify themes. The narratives were then used to create core composite narratives between those who changed career intentions versus those who did not. These core composite narratives were compared through narrative analysis to explore differences in how students appraised chance events.

action, climax, falling action, and resolution [33] that aligned with the Appraisal Theory components (Table 2). For example, the rising action of a story arc is the occurrence of the chance event, and the climax of the story are the primary and secondary appraisals individuals make. Aligning the story arc components and appraisal processes guided our placement of information from the transcripts into a narrative that chronologically tracked the emotional and cognitive processing the student described over time (see S2 Appendix, for an example of a story arc of a participant aligned with Appraisal Theory). This narrative structure will henceforth be referred to as the Appraisal Theory Story Arc (ATSA), which focuses on key elements of the appraisal process (e.g., primary appraisal, secondary appraisal, and reappraisal) within the context of each participant's story. Each graduate student's ATSA was created by H.F. and reviewed by E.S. to ensure they reflected the information in the transcripts.

While Appraisal Theory provided a structure for organizing the participants' stories, it was not used deductively to identify themes. The thematic analysis was instead inductive, meaning the themes were derived from the data itself, rather than being based on pre-existing frameworks [34,35]. To minimize the influence of Appraisal Theory, H.F. employed a bracketing method that involved explicitly documenting her knowledge of Appraisal Theory, approaching the data with a focus on the participants' words and experiences, coding only what was explicitly present in the transcripts [36].

**Table 2. Mapping Story Arc to Appraisal Theory.**

| Story Arc Component | Appraisal Theory Component | Description |
|---|---|---|
| **Exposition** (Characters/Setting) | Participant Context Before Event | Original goals and interests before the chance event occurred. |
| **Rising Action** (Series of events) | Chance Event | Details about the chance event. |
| **Climax** (Turning point) | Primary & Secondary Appraisal | Immediate emotional and cognitive responses. |
| **Falling Action** (Movement to a new normal) | Reappraisal | Reflection of the event after time has passed. |
| **Resolution** | Outcome | Overall perception of event and its impact on their career intentions. |

To conduct thematic analysis, the narratives of each participant were reviewed by H.F. and codes were identified by grouping similar ideas into a category and description. The coding was both holistic and chronological, looking for similar ideas that occurred for participants who were at the same stage of the ATSA. After codes were identified, they were grouped into broader themes. Both authors then assessed the themes by reviewing the data to ensure that the themes represented the participants' experiences accurately [37]. The theme names and descriptions were also reviewed by three other education researchers to check that the descriptions and titles sufficiently conveyed the findings.

**Research question 2: How did chance event appraisals differ among the graduate student participants who changed their career intention versus those who did not?** Participants' ATSAs were compiled into two groups to create two synthesized narratives called "core composite narratives;" one was for participants who changed their career intentions since the start of graduate school, and one was for those who did not. A core composite narrative is a synthesized story that combines the individual stories of participants, highlighting common themes and ideas from that group [38,39]. Once these two composite narratives were created, the prevalence of the themes from research question 1 were identified in each core narrative. This allowed us to identify differences between these groups in how their chance events were appraised and reappraised, and how this potentially related to their career intentions.

## Results

### Research question 1: How did life science graduate students appraise and reappraise chance events?

Seven themes were identified that reflected how the participants experienced their chance events: *Initial Stressors, Initial Emotional Toll, Help-seeking, Self-reliance, Realizations, Resilience, and Growth* (Fig 3). These themes will be discussed below within the context of the Appraisal Theory Story Arc (ATSA).

**Initial Stressors.** The theme *Initial Stressors* emerged in participants' narratives when the interviewer asked them to describe what was happening in their program before they encountered the chance event. This represented the

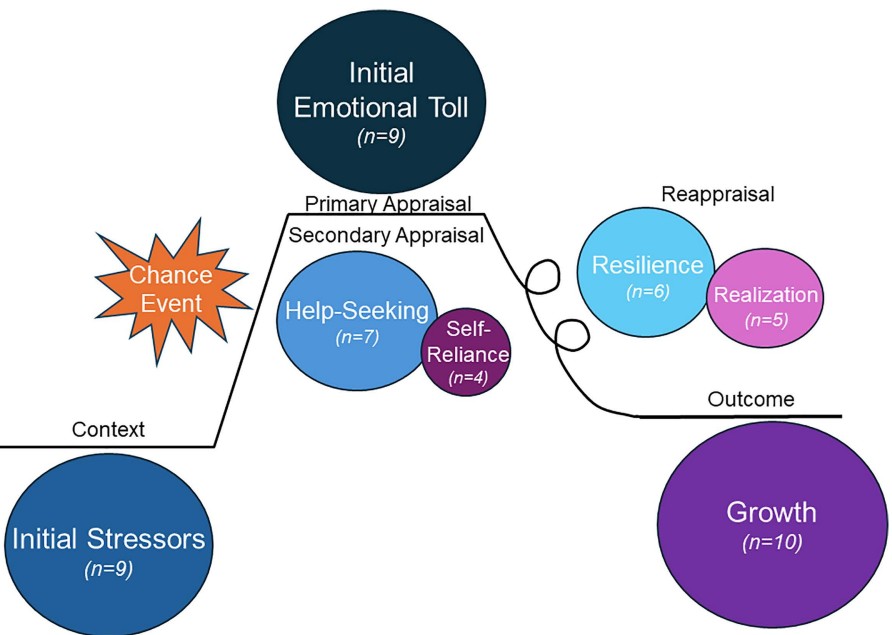

**Fig 3. Bubble chart of theme prevalence mapped onto Appraisal Theory Story Arc.** Themes are shown in circles with the size of each circle corresponding to the number of participants (number out of 10) who mentioned the theme.

"context" stage of ATSA (i.e., exposition). Nine participants expressed challenges, obstacles, or stressors they were already experiencing before something unexpected happened, which shaped how they processed and responded to the chance events. Their narratives described contextual factors such as struggling with confidence and self-doubt, financial constraints or limited job opportunities, or dealing with academic expectations and pressures. Yellow, for example, began his story by explaining his doubt about his career path, choosing between an MD or a PhD, saying:

*"I was filling out grad school applications and really getting down to nitty gritty stuff as COVID started to be found in China… there was a large moment of …if I'm going to be [researching] am I better served studying medical (topics)?..."*

In Yellow's narrative, he experienced self-doubt and internal conflict about the career path that would have the most impact on society. With the chance event of COVID, this context of wanting to make a professional impact made them question their choice to not take a medical focus for their career. Notably, the challenges or stressors described within this theme were not unplanned or unpredictable but were often acknowledged as expected difficulties inherent to navigating academic and professional journeys. For example, Coffee stated,

*"…I tried to set up a new study system and experience(d) all the teething troubles that come with that…I thought I had sort of been through that phase of like, trying and failing and trying and failing…"*

Coffee's comment about "trying and failing" over time was framed as a normal struggle of graduate study, but the unexpected barriers created by COVID-19 added more challenges to those already difficult tasks.

**Initial Emotional Toll.**  When asked to reflect on and describe what they were feeling when the chance event first happened (e.g., primary appraisal), the *Initial Emotional Toll* theme emerged among nine of the participants. They described the event as unexpected, leading to negative emotions like burnout, frustration, fear, and self-doubt, highlighting the significant negative impact the event had on them at that moment. These negative emotions were particularly directed toward uncertainty related to their research or their academic responsibilities. For example, in response to unexpected periods of difficulty concentrating and breathing, Violet stated,

*"I was just really scared. Because I thought that like something physically was like wrong with me. And I was really afraid of not being able to finish grad school or be able to keep up with my classmates…"*

Violet's reflections showed how deeply these unexpected occurrences could affect one's self-confidence about her future. While the initial emotional toll was often immediate, some participants found that their emotional responses evolved, in some cases intensifying as they realized the full extent of the situation. For instance, Olive described how his initial negative viewpoint regarding the COVID-19 pandemic shifted to become even more negative as he began to see the effects of his work, stating,

*"COVID Definitely, brought some negative emotions to field work… [I felt] pretty shitty. Like just pretty negative overall and well…as it impacted my research and impacted me more and more, it got more and more negative… "*

These reflections showed that the event itself created an initial emotional shock, but its impacts lingered and evolved over time for many. Given the longevity of the emotional responses, participants often utilized strategies to navigate these feelings, such as help-seeking and self-reliant behaviors.

**Help-Seeking.**  As participants cognitively processed the chance event (secondary appraisal), the *Help-Seeking* theme reflected descriptions of one common type of coping strategy. *Help-seeking* involved asking for guidance or support from others within the student's network, which included family, peers, faculty, or advisors. Seven participants

described seeking this help when navigating the emotional impact of the chance event. For example, in response to being unexpectedly diagnosed with a panic disorder, Violet said,

> "I began kind of reaching out for help…I had a long discussion with my advisor. And I was like…this is going on, I'm not doing well. And he was like, it was really a lot. All my friends and family that were like, give yourself grace…"

In going to others for support, these students were able to receive validation and resources to cope with the challenges of the unexpected event. Three participants, however, either delayed seeking help from their advisor or chose not to seek their advisor's guidance at all when faced with the unexpected. For example, Lime was struggling with feelings of burnout with research and decided to look up teaching positions. He found a teaching position and was excited about the potential job opportunity. While Lime talked to his peers and was encouraged to apply, he felt that he could not seek guidance from his advisor. Lime stated in his narrative,

> "In hindsight, it would have been better for me to talk to my advisor about the opportunity. But I kind of suspected what the feeling would have been… My advisor would have said, "you don't need to be doing that, you need to stay and get your PhD done. Focus on doing research" … It was hard for me to not put him down as a reference…"

Although Lime chose not to seek guidance from his advisor, he still turned to his peers for support, choosing those he knew would be encouraging and avoiding those he perceived would not support his choices. Thus, Help-seeking did vary among the participants in terms of who they spoke with.

There were also two instances where participants (Bronze and Pearl) reached out to their advisors but felt unsupported or found the advice unhelpful. For example, Pearl described seeking support from a faculty member in response to unexpected negative feedback from students. She explained,

> "I did [talk to the instructor]... the instructor obviously wasn't as hurt by what the student was saying as I was… And I think they thought it wasn't a major issue. I certainly think they understood I was upset. But it was hard to communicate what I was feeling."

Pearl's experience highlights challenges participants may face when revealing events that have caused them emotional struggles that faculty minimize or don't understand, potentially creating a reluctance to disclose additional events or feelings in the future.

Seeking help was not the only coping strategy mentioned by participants. Some instead spoke about relying on their own judgment and resources.

**Self-Reliance.** The second coping strategy that aligned with secondary appraisal and voiced by some participants was Self-Reliance. This theme encapsulates a participant's reliance on their own capabilities, judgment, skills, and resources to navigate the chance event. Four participants demonstrated this theme by not seeking help from others but instead engaging in self-talk, conducting personal research, or making decisions independently. For example, after being told her advisor was unexpectedly moving to another university, Khaki needed to decide if she would move with her advisor or stay at her current university. Khaki stated,

> "…it only took me about five hours to decide I was moving. And the ultimate reason for that was, I knew this would be better for my career in the long run if I did go… I'm thinking anything from like, people that I could potentially do postdocs with…or help me with my current research, or…people that could…help get me jobs in industry in the future …"

Khaki independently and quickly evaluated her options and made decisions that she perceived would benefit her career goals. Other participants engaged in behaviors like self-talk or reliance on their skills. For example, in response to the unexpected impacts on his research due to the COVID-19 pandemic, Olive stated,

*"I was more isolated to address a lot of underlying doubts and underlying confidence issues…it forced me to you know, really take them head on and push through…when I was crying in the field I'd be like "alright [Olive], you're a bad bitch like get back to it…"*

Olive used self-talk as a way to encourage himself and maintain motivation in light of the challenges he faced both in response to the pandemic and research, showing a reliance on internal strategies versus social help-seeking. By utilizing coping strategies participants were able to navigate through the impacts of the unexpected event, and this process seemed to take them through a reconciliation about how they felt about it.

**Realizations.** The *Realization* theme was created from participant descriptions of reassessing and reflecting on their experiences during the reappraisal stage of ATSA. This theme captures participant stories about how they gained new personal insights, recognized what was truly important to them, and realigned their interests and career paths accordingly. Five participants described these realizations when asked how they felt about the chance event a few months after it happened. For example, Violet, whose chance event was an unexpected diagnosis of a panic disorder, decided to leave her program with a master's degree. Violet stated:

*"I think that I felt like going to grad school I had to like prove myself and go get a PhD and get a really good job. And I realized through all this that taking care of your body, and your mental health is wildly important…I just want to be happy and have a chill life and I'm okay with not getting a PhD and leave it with Masters."*

This suggests that Violet used the chance event as a moment to shift her focus from external validation and career expectations to prioritizing her mental and physical health. Another participant, Beige, described that after choosing to act on an unexpected job opening, she recognized her potential options. She stated,

*"…even though I didn't get that specific job, I'm glad that I applied for it. And I'm glad that I found a job that sort of made me open my eyes and realize the things that I really enjoyed doing, not things that I think I should enjoy, or things that I'm supposed to enjoy based on the fact that I'm in grad school."*

Her chance opportunity led to a realignment of her career goals with her interests, marking a significant and personal discovery for Beige. The remaining three participants—Aqua, Coffee, and Lime—also shared similar realizations of what would make them happy and fulfilled—whether that was meeting personal values or goals or professional ones—suggesting that their chance events were pivotal moments in their career paths. In contrast, five participants had a different adjustment: Resilience.

**Resilience.** The *Resilience* theme was also created from participant descriptions of reassessing and reflecting on their experiences during the reappraisal stage of ATSA. In this case, participants used the chance events to reaffirm their direction, push through setbacks, and continue to progress despite difficulties. Their stories often relayed a commitment to persevere and an ability to adapt and navigate challenges. For example, after unexpectedly needing to change projects in light of the COVID-19 pandemic, Bronze stated in her narrative,

*"I was like, gosh, I hate wet lab work. But…it showed me that I had grit that I could get things done. Even if it wasn't something I was super interested in…And I feel like it made me more confident… And I feel like it also showed them I was able to do the project…"*

This reflection from Bronze highlighted her ability to adapt, persist, and increase her confidence without changing her career intentions. Other participants' narratives in this group demonstrated similar stories of resilience in the face of the challenges of chance events. Sometimes this resilience sounded like an acceptance of their current situation, but it was still not about altering their existing path. For example, Olive stated,

> *"I'm just like currently, kind of indifferent and neutral to it…this is just a new normal and I gotta push through and get through it…"*

Olive talked about his work after a chance event as a "new normal," which showed that there did not need to be a change in behavior or view of the situation. This suggests that resilience does not always involve significant shifts in perspective. Instead, it can appear as acceptance and determination to keep moving forward despite challenges.

**Growth.** The final stage of ATSA involves participants reflecting on their chance events, and participant reflections led to the creation of the *Growth* theme. This theme described participant feelings of increased confidence, acceptance of personal struggles, and improved skill sets that made the chance event a transformative experience for them. For example, Khaki reflected on new skills she needed to learn, the leadership role she took, and the professional connections she made in moving to a new university with her advisor, and how all had a major influence on how she now views the chance event. Khaki stated,

> *"I think it helped my confidence a lot…I'm able to see myself in situations I would not see myself in before looking at postdoc opportunities…now I feel like I have the confidence to be like, Okay, I can…teach myself how to do that."*

Other participants described building skills in self-advocacy that supported their well-being. For example, Beige shared that after going to counseling and taking medication for anxiety and depression in response to her unexpected isolation during the COVID-19 pandemic, she felt like she was better equipped to communicate with her committee. She stated,

> *"I do feel now that I'm more able to speak up about things that are making me uncomfortable, or situations where I'm unhappy and be able to act on things more, rather than just wallowing in my emotions."*

The reflections of participants about their growth highlight a development of self-awareness and confidence that may emerge from dealing with challenging emotions and situations that arise from unexpected events.

**Research question 2: How did the narratives of chance event appraisals differ among participants who changed their career intentions versus those who did not?**

The core composite narratives for the participants who changed their career intentions and those who did not are shown in S3 Appendix. The themes from the individual narratives were synthesized into two composite narratives to identify similarities and potential differences in how each group reacted to the chance events (Fig 4). This analysis revealed that the themes of *Initial Stressors, Initial Emotional Toll,* and *Growth* were shared across both core composite narratives. However, the *Help-Seeking, Self-Reliance, Realizations,* and *Resilience* themes differed in prevalence among participants who did or did not change their career intentions. These similarities and differences are explained below.

All of the participants were experiencing negative feelings or emotional strain about their work before their chance event (i.e., *Initial Stressors*). Participants viewed these challenges as an expected part of graduate school and something they needed to push through to reach their goal. Chance events then created Initial Emotional Toll whether participants changed their careers or not; this added additional strain to their graduate program. All participants used coping strategies to manage these emotions and, over time, reappraised the chance events more positively by confronting their initial stressors. Ultimately, participants in both groups believed chance events were transformative

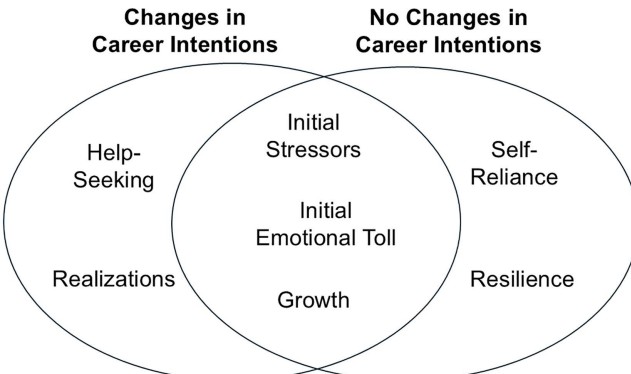

**Fig 4. Venn diagram of themes present among participants in composite narratives.** Common themes across all participants include initial stressors, initial emotional toll, and growth. For those who changed their career intentions in graduate school, the distinct themes were help-seeking and realizations. In contrast, participants who did not change their career intentions exhibited themes of self-reliance and resilience.

and helped them develop as individuals, gaining insights and skills that contributed to their personal or professional growth (i.e., *Growth*).

Despite these similarities, notable differences emerged between the core composite narratives when examining how each group responded to chance events. Participants who changed their career intentions talked about seeking external support and advice from peers, and later from advisors and faculty (i.e., *Help-Seeking*). In contrast, participants who did not change their career intentions talked more about relying on internal strategies such as self-reflection and self-talk, demonstrating S*elf-reliance* in coping with their challenges. The reasons for positive reappraisals also differed between the two core composite narratives. For those who changed their career intentions, they talked about the chance event as a catalyst for self-discovery, helping them reassess their values, interests, and well-being (i.e., *Realizations*). Conversely, participants who did not change their career intentions discussed chance events as opportunities to build skill sets and confidence in their existing career paths (i.e., *Resilience*). Thus, regardless of chance event type, it was the way they reacted to the unexpected event that seemed to be more aligned with any career intention changes or not.

## Discussion

This study explored life science graduate student narratives about their emotional and cognitive appraisals and reappraisals of chance events, revealing story arcs that aligned with Appraisal Theory. After participants described the context of the chance event (i.e., exposition and rising actions), many expressed an *Initial Emotional Toll* consistent with primary appraisal and the climax of a story. During secondary appraisal, they talked about evaluating resources and methods to cope *(Help-seeking or Self-reliance)*. The reappraisal stage (i.e., falling action of a story) marked where many participants reinterpreted the event and fostered *Realizations* (new career intentions) or *Resilience* (maintaining their career intention) that often led to professional or personal *Growth,* regardless of whether their career intentions changed or not. These findings support the value of using Appraisal Theory to study the emotional and cognitive processing of impactful experiences of graduate students, as explored by [21], with this study expanding the analysis to secondary appraisals and reappraisals and how it may relate to shifts in career intentions. Importantly, the Appraisal Theory Story Arcs (ATSAs) in this study suggested a transformative arc from negative to more positive perceptions of chance events over time and hinted at how they can be leveraged to enhance career reflection.

Comparing narratives of participants who changed their career intentions with those who did not suggests initial and final stages of the ATSAs were thematically similar, while the middle climax and falling action stages (i.e.,

primary and secondary appraisals and reappraisals) thematically differed. Those who changed their career intentions seemed to cope with their challenges by relying on external support, while those who did not change career intentions seemed to rely on internal strategies like self-reliance and resilience. Those who changed their career intentions often talked about aligning their new career intention with their well-being and interests. In contrast, those who did not change their career intentions often mentioned gaining skills that reinforced their existing career intentions. These results align with the findings of Lipshits-Brazier and colleagues [40] who found that productive coping styles (e.g., information seeking, problem solving, flexibility in decision-making, and self-regulation) were associated with advanced career decision making and higher levels of career choice satisfaction among undergraduates. While Lipshits-Brazier et al.'s study [40] focused on undergraduates, our work also hints that productive coping styles may shape careers in ways that are positive, regardless of whether students change career intentions or not. While recommendations for generalizable coping strategies seem unlikely given the number of factors related to career intentions, the value of coping skills for career decision-making is a topic that should be further explored for graduate students.

This study also served as a critical reminder that career intention shifts can be positive outcomes for graduate students despite often being cited in the literature as a negative outcome [41,42]. We found that all participants in this study were satisfied with their choices in response to the chance event regardless of whether they shifted their career intentions or not; all participants also saw the event as an important catalyst for reflection about their career path. Although this may have been a peculiarity of this graduate student sample, we contend that mentors, advisors, faculty, peers, and other career support services should recognize the potential of chance events to be a healthy part of career exploration. It is also a reminder that graduate students need both emotional and cognitive support as they navigate unexpected events and examine their implications for their careers. Much of the career development literature highlighted skill development, networking opportunities, career planning, or the provision of resources, but there is less of a focus on the types of coping skills that support the emotional needs of graduate students when they encounter unexpected events in graduate school. Ryan et al. [43], for example, identified gaps in supporting graduate students in their career development and recommended several different ways to address them which included improving and expanding job resources, enhancing emotional-psychological-social support, adopting a whole-university approach to well-being, and fostering supportive and inclusive research cultures. Thus, our study adds to a call to improve the types of support provided to graduate students as they navigate graduate school.

## Stress and chance events

The findings of this study suggest the stress and emotional challenges that chance events can create for graduate students, regardless of career intention shifts. This emotional toll was a shared experience among our participants, often creating uncertainty that led to increased stress and anxiety. Higher education is already a demanding environment filled with prolonged stress and high anxiety levels that students must manage [6,44]. This study adds to the literature by suggesting that chance events may be an additional and different source of stress that graduate students must manage on top of the expected challenges of graduate study. Students, faculty, and departments should all work together to create and foster supportive academic environments that support stress from both expected and unexpected sources. Students can encourage open conversations with other students, providing peer support, and advocate for more structured mental health and well-being resources. Faculty and departments should be aware that students may be experiencing the extra burdens of unexpected events. In a similar vein, Ma et al. [45] suggest normalizing meetings with students that ask about their needs and overall well-being. This would allow faculty to identify student struggles and offer timely support, such as connecting students to resources such as mental health services or support groups. Although this type of anxiety and stress support has not been typical in graduate school, rising rates of mental health struggles [46,47] suggest that now may be the time to establish new norms of care in programs.

## Coping and chance events

Coping strategies play a crucial role in helping individuals manage stress and anxiety to maintain positive mental health [20]. Musgrove et al. [48] interviewed graduate students about the anxiety they experienced conducting research and teaching and suggested that coping strategies like problem-solving, emotion regulation/self-reliance, help-seeking, and cognitive restructuring are adaptive strategies that can reduce graduate student anxiety. In their study, graduate students used adaptive coping strategies more often than maladaptive coping strategies [48]. Students in our study also mainly discussed using adaptive coping strategies, choosing to use techniques that helped them process and manage the emotion created by a chance event versus ignoring it. This active engagement with the emotional processing of the event seemed to allow them to consider the chance event's impacts on their career intentions. It is difficult to determine generalizability with a small sample of participants who were all willing to talk about their experiences; however, more research needs to be done to ascertain the scope of coping strategies that graduate students use to manage the emotional impacts of chance events. Understanding why students might choose different coping strategies and how these choices influence their reappraisal processes could provide valuable insights into the dynamics of stress management and decision-making within graduate education.

Help-seeking coping strategies were prominent in the narratives of those who shifted their career intentions, but their effectiveness seemed to vary. Pearl, for example, found little support when discussing the unexpected negative feedback from students with her advisor. Her advisor's recommendation to "wait and reflect" inadvertently encouraged rumination, a maladaptive coping strategy [48]. Although help-seeking is cited as an effective way to problem-solve and enhance academic performance [49,50], this study reminds us that not all instances of help-seeking are beneficial, especially when the quality of support differs across faculty. Additionally, participants sometimes initially sought guidance from peers or family and waited—or avoided—seeking help from faculty. Lime avoided discussing a job opportunity with his advisor, anticipating a lack of support. This is consistent with research that indicates students may prefer informal social networks to avoid perceived threats or negative judgment, particularly from faculty [50–52]. This finding underscores the importance of creating a supportive environment where students feel safe discussing changes in their career intentions. As increasing numbers of graduate students seek jobs outside of academia [3,4,53], students, faculty, and departments should work together to normalize and support career changes by fostering open communication and development of career resources.

## Growth from chance events

Initially, most participants viewed chance events as highly stressful and emotionally disruptive, yet all participants ended up reappraising these events as more positive over time. This finding aligns with the work of Lazarus and Folkman [18], who frame reappraisals as a form of meaning-based coping where individuals reinterpret stressful events as valuable or beneficial. Yet our study indicated that the nature of these reappraisals differed among those who changed versus those who did not change their career intentions while in their programs. Those who changed career intentions often described realizations that led them to realign their goals and prioritize their well-being. Conversely, participants who maintained their career intentions seemed to view chance events as opportunities to build resilience and enhance their skills, which reinforced their confidence and ability to overcome obstacles in their career path. Each reappraised the stressors in ways that aligned with their personal goals and values [18,54]. This leads to the question of whether those who changed their career intentions were initially more misaligned in their values, goals, and careers than the other participants, making them more willing to re-examine and adjust their goals than the others. Future studies should investigate whether some students might have a higher malleability of goals and careers, and if so, why.

## Limitations and future work

There were several factors related to the study that limit the generalizability of our findings. First, this study was conducted during the COVID-19 pandemic. Several of the chance events mentioned by graduate students were related to

the pandemic which was an unusual historical event. The isolation and physical distancing likely exacerbated feelings of stress and anxiety, and made connections with advisors, faculty, and peers difficult. This may have explained some of the findings related to help-seeking where students did not go to mentors. However, this highlights the importance of ensuring graduate students feel connected and supported while in their graduate programs.

Additionally, our results were likely impacted by response and recall bias which limits the applicability of these findings to broader groups of graduate students. The sample was relatively small and limited to life science graduate students at one institution. It is also likely that those who chose to be interviewed felt they had a positive story to share, resulting in stories with positive endings. These participants were also selected because they indicated in the survey that they had had a chance event that highly impacted their career. Thus, their stories of personal and professional growth are likely not generalizable. This study also relied on retrospective accounts which may influence how participants remembered and described their experiences. However, even if their recall is not perfect, their narrative is driving their current behavior, which makes it a salient experience to study. Thus, participants' stories may not fully capture the complexity of their emotional and cognitive responses. Future work can focus on recruiting a larger sample across academic disciplines and examine differences among students at different stages in their programs.

It is important to note that not all chance events may be equal in terms of influence on careers and/or progress. For example, one participant may experience two separate chance events, but feel that one event was more significant to them. Additionally, two participants may experience the same type of event and perceive the significance differently, and then respond differently. It is possible that these differences depend on timing of the event, the individual's personal context, and the meaning the individual assigns to the event. Since this study only focused on one highly impactful event for each participant, we do not know how the different experiences from each event may have influenced career decisions and progress. Future work can examine how students make sense of their experiences with multiple chance events over time and whether certain types of events may be more influential in career intention shifts.

Lastly, while Appraisal Theory provided a useful lens for examining the emotional and cognitive processing of graduate students experiencing a highly impactful chance event, this theory prioritizes how individuals make meaning of events and do not explicitly account for broader institutional or structural contexts. These contextual factors certainly influence and shape appraisals of events, but our focus was not on these factors. In this study, we did not examine participants' previous experiences or academic contexts, which likely did influence their responses. Therefore, future work should incorporate different theories or frameworks that more directly account for these broader contexts. For example, Self-Determination Theory (SDT) posits that variations in students' autonomy (i.e., locus of control for behaviors and goals), competence (i.e., efficacy to learn skills and master tasks), and relatedness (i.e., sense of belonging) could influence how they reinterpret and respond to these events [55]. The Planned Happenstance Learning Theory (PHLT) emphasizes skills such as curiosity, persistence, optimism, risk-taking, and flexibility that may help students to effectively leverage chance events [56,57]. Ultimately, these theories may provide a foundation for examining the interaction of personal and contextual factors that influence how students navigate chance events, which in turn, may provide insights into improved career development resources for these students.

## Conclusions

This study analyzed narratives about how graduate students emotionally and cognitively processed chance events, and how these processes inspired personal and professional growth in many of the students. Many chance events were initially seen as negative but were transformed into positive catalysts for career development, regardless of whether the students changed their career intentions or not, although how each group processed the events appeared to slightly differ. These findings emphasize the importance of student reflection and using adaptive coping strategies when navigating, interpreting, and reassessing chance events. This study also suggests that impactful chance events are prevalent in graduate education and calls for faculty and departments to support graduate students by helping them use unexpected moments as opportunities for personal reflection and growth.

## Supporting information

**S1 Appendix. Table of interview questions for the study.**
(PDF)

**S2 Appendix. Example of Appraisal Theory story arc.**
(PDF)

**S3 Appendix. Core composite narratives.**
(PDF)

**S4 Appendix. De-identified data.**
(PPTX)

## Acknowledgments

Many thanks to Dr. Jessica Budke for her valuable feedback on this document. I am also deeply thankful to my lab mates, Dr. Maryrose Weatherton and Bailey Von der Mehden, for their continuous support and insights during this project. Two reviewers provided comments that improved the manuscript. Lastly, I would like to thank the participants in this study; their willingness to share their experiences was instrumental to this work.

## Author contributions

**Conceptualization:** Hope Ferguson, Elisabeth E. Schussler.

**Data curation:** Hope Ferguson.

**Formal analysis:** Hope Ferguson.

**Investigation:** Hope Ferguson.

**Methodology:** Hope Ferguson, Elisabeth E. Schussler.

**Project administration:** Hope Ferguson.

**Resources:** Elisabeth E. Schussler.

**Supervision:** Elisabeth E. Schussler.

**Validation:** Hope Ferguson, Elisabeth E. Schussler.

**Visualization:** Hope Ferguson.

**Writing – original draft:** Hope Ferguson, Elisabeth E. Schussler.

**Writing – review & editing:** Hope Ferguson, Elisabeth E. Schussler.

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
