## [Decision Letter · Decision Letter 0]

2 May 2025

Dear Dr. Ferguson,

Thank you for submitting your manuscript to PLOS ONE. After careful consideration, we feel that it has merit but does not fully meet PLOS ONE’s publication criteria as it currently stands. Therefore, we invite you to submit a revised version of the manuscript that addresses the points raised during the review process.

We look forward to receiving your revised manuscript.

Kind regards,

Patricia Anne Morris

Academic Editor

PLOS ONE

2. In the online submission form, you indicated that [The interview transcripts generated and analyzed during the current study are not

publicly available because they contain data that allows identification of the participants.

A limited form of the data can be made available from the corresponding author on

reasonable request.].

Additional Editor Comments:

Dear Authors,

Thank you for the opportunity to review this paper. Overall I find the topic quite interesting, and the writing is sound. Both reviewers speak to the need for clarity in the theoretical framework and for contextualizing/clarifying the concept of chance events. Reviewer One questions whether the theoretical framework you have used is appropriate for the analysis. Reviewer Two suggests there are issues with the concepts of chance events and career aspirations/intentions/trajectories. This revision will require significant reworking to address the reviewers' concerns.

Thank you for your submission.

best,

Patricia Morris

Reviewers' comments:

Reviewer's Responses to Questions

**Comments to the Author**

1. Is the manuscript technically sound, and do the data support the conclusions?

Reviewer #1: Yes

Reviewer #2: Partly

2. Has the statistical analysis been performed appropriately and rigorously?

Reviewer #1: N/A

Reviewer #2: N/A

3. Have the authors made all data underlying the findings in their manuscript fully available?

Reviewer #1: Yes

Reviewer #2: Yes

4. Is the manuscript presented in an intelligible fashion and written in standard English?

Reviewer #1: Yes

Reviewer #2: Yes

Reviewer #1: General comments: Thank you for the opportunity to review the paper entitled: Processing the Unexpected: A Qualitative Study of Graduate Student Appraisals of Chance Events

This paper was interesting. Overall, I would recommend this manuscript for publication with major revisions, as the methodology needs to be clarified. Please see below for the comments related to this paper.

TITLE

• The title is appropriate and reflects the content of the manuscript.

ABSTRACT

• The abstract is generally well written.

INTRODUCTION

• The introduction is generally well written and leads to the aim of the study.

• Lines 73–75: The authors state that nearly all participants recall an event that may have influenced their career choice. While this is certainly noteworthy, the assertion appears somewhat reductive. The degree to which such an event may have influenced career trajectories can vary significantly across individuals. It would be important to highlight that it is not merely the presence or absence of such an event that matters, but also the context in which it occurred, the extent of its impact, and whether it triggered a cascade of subsequent chance events. Furthermore, the nature of the chance event itself is not discussed. For instance, a chance event such as receiving unexpected funding might have a substantially different impact compared to meeting a particular person. This section would benefit from a more nuanced and in-depth discussion of these aspects.

• Lines 79–81: Similarly, this passage does not offer insight into the broader context or the lasting impacts of the events described. The argument could be strengthened by elaborating on how contextual elements mediate or moderate the influence of these events.

• Lines 81–83: While I agree with the authors on the importance of better understanding these processes, the discussion does not sufficiently emphasize the contextual factors in which emotional and cognitive responses occur. The focus seems to rest primarily on the individual's internal responses, with limited consideration of environmental factors (e.g., physical, economic, social) that may shape or constrain those responses.

• Lines 93–95: This section emphasizes cognitive processes, which is relevant. However, it would be valuable to incorporate a discussion of how individuals assess both internal resources (e.g., capacity to tolerate uncertainty) and external resources (e.g., availability of social support). A more holistic perspective could enhance the theoretical depth of the analysis.

THEORETICAL FRAMEWORK

• Lines 110–111: The authors highlight that the interpretation of events varies according to personal perceptions. While this point is valid, it may conflate individual perception with the broader context. An individual may indeed interpret an event in a particular way, but this perception does not negate the influence of the stable or shared context in which the event occurs. A clearer distinction between individual perception and contextual conditions would strengthen this argument.

• Lines 117–118: The mention of “available resources” appears here for the first time, without any prior discussion in the introduction. Given the centrality of available resources in shaping individuals' interpretations of chance events, it would be important to revise the introduction to incorporate a more explicit discussion of their role and implications.

• Lines 153–154: This statement appears to partially repeat the content presented in lines 98–102. To enhance clarity and avoid redundancy, I would suggest retaining only the formulation in lines 153–154, which is more concise and better integrated.

• Lines 159–160: The number of graduate students interviewed is mentioned in this section, yet this detail directly pertains to the research questions and design. As such, it would be more appropriate to present this information later in the methods section, rather than here.

METHODS

• Overall: The methodology section is generally well-structured and clearly written. However, several key clarifications are necessary to enhance transparency and rigor.

• COREQ Checklist: The manuscript would benefit from explicitly referencing and aligning with the COREQ (Consolidated Criteria for Reporting Qualitative Research) checklist. This would help ensure completeness and adherence to established standards for qualitative research reporting.

• Line 169: The term “life science graduates” is introduced without a prior definition or justification. It would be important to clarify why this specific population was selected. What makes their experiences with chance events distinct compared to other graduate students? This choice raises questions regarding the transferability of findings to other academic fields.

• Contextualization of Chance Events: The discussion of chance events lacks historical, socioeconomic, and sociocultural contextualization. Situating the participants within such broader contexts would provide readers with a more nuanced understanding of how these events are experienced and interpreted.

• Line 186: The manuscript states that “five individuals from this group” were selected, but the selection criteria and process are not described. Clarifying how participants were chosen would improve transparency and allow readers to better assess potential biases.

• Lines 194–195: The authors state that “verbal consent was appropriate for this work.” This raises concerns, particularly as participants were initially recruited following a survey. It is unclear whether participants were provided with a full explanation of their rights and the scope of the qualitative interviews. There is a distinction between simply obtaining verbal consent (e.g., a “yes”) and securing informed verbal consent based on a detailed explanation of risks, benefits, and participant rights. This point requires further elaboration and ethical justification.

• Line 199: Both master’s and PhD students were included in the study, yet these programs can differ greatly in duration, structure, and disciplinary culture. The rationale behind including such heterogeneity in participants’ academic backgrounds should be clarified. What are the implications of this choice for the interpretation of the results?

• Lines 199–211: Much of the content in this section appears to describe findings or characteristics of the participants and may be more appropriately placed in the results section rather than the methods.

• Lines 201–203: The sociodemographic profile of the participants appears quite homogeneous. This raises important questions about the diversity of experiences captured in the study. Were any deliberate efforts made to recruit participants outside the 25–34 age range? The contexts and chance events experienced by individuals in different life stages may vary significantly.

• Lines 215–218: It is unclear why the interview guide was not directly informed by the theoretical framework presented earlier in the manuscript. Instead, the authors indicate it was developed based on prior survey responses. A stronger justification is needed here, particularly regarding the relevance and limitations of this approach.

• Lines 220–223: It is noted that interview questions were developed using participants’ own survey responses. This raises ethical concerns: Did participants explicitly consent to this kind of data linkage and use? The overlap between the survey and interview phases may require additional ethical safeguards and greater transparency in the consent process.

• Line 234: The manuscript refers to HF’s role in the research, but does not sufficiently describe HF’s experience with conducting qualitative research. This information is relevant for assessing the credibility of data collection and analysis and should be expanded.

• Lines 244–248: I appreciate the authors’ transparency regarding the presence of a peer who conducted the interviews. However, this presence may have introduced social desirability bias, potentially influencing participants’ responses. In future studies, it might be advisable to involve an independent interviewer to minimize such bias.

• Use of Reflexivity Tools: There is no mention of the use of reflexive tools such as a research journal or analytic memos. Incorporating these tools is a widely recommended practice in qualitative research to support transparency and rigor, particularly in the interpretation of data.

RESULTS

• Overall: The results section presents compelling and relevant findings; however, it would benefit from further synthesis and conciseness. At 14 pages, it currently feels overly lengthy, which may hinder readability and reduce the clarity of the main insights.

• Lines 409–466: This theme includes a particularly large number of direct quotations, many of which are quite long. In some cases, the accompanying analytical commentary is relatively brief (e.g., lines 455–457 and 465–466), making it difficult to fully grasp the authors’ interpretive stance. I would recommend selecting fewer, more illustrative quotes and expanding the analytical reflections that follow, to better highlight the significance of the excerpts in relation to the theme.

DISCUSSION

• Overall: The discussion is engaging, well-structured, and thoughtfully written.

• Lines 763–781: The content in this section, while relevant, might be more effectively integrated throughout the discussion rather than presented as a separate subsection. Embedding these insights into the broader analytical narrative could enhance the coherence and flow of the argument.

• Lines 792–800: The limitation mentioned regarding the interviewer’s position (i.e., HF being a PhD student at the same institution as some participants) warrants a more in-depth examination. This dynamic may have influenced both the interview process and subsequent data interpretation, potentially introducing bias or social desirability effects. Stronger justification or mitigation strategies should be provided to address these concerns transparently.

• Participant-Specific Limitations: The limitations section places minimal emphasis on the specificity of the sample—namely, life science students. However, it is plausible that the experiences of this group differ from those in other graduate fields (e.g., health sciences, humanities, engineering). The authors should more clearly situate their findings within the context of this specific population and acknowledge the limited generalizability of results due to the narrow disciplinary focus.

CONCLUSION

• The conclusion is adequate.

Reviewer #2: Thank you for the opportunity to review manuscript PONE-D-25-11301 titled “Processing the Unexpected: A Qualitative Study of Graduate Student Appraisals of Chance Events.” The topic of stressors and needs for supports for students enrolled in post-secondary education is timely and of great interest to many. Overall, the manuscript is well written and organized, the appendices and supplementary files are informative and well done - however revisions are required.

• I was unfamiliar with the term “chance events” within the context of graduate studies. It likely explains why I was not sure what the study was about until I began reading the manuscript. Because of this, I wonder if the authors should reconsider the title of the paper so its relevance is apparent to those interested in supporting graduate students.

• A major issue I have with the manuscript pertains to its focus on students’ careers. At times the authors discuss students’ careers, while other times it is career intentions, career trajectories, and career outcomes. Given participants were enrolled in either master or PhD graduate programs, it appears the only thing that would be relevant is career intentions – and maybe academic goals. The authors are encouraged to carefully consider what the main concept of interest is. In keeping with this, line 389 seems to suggest participants may have been focusing on their career goals (which is different than academic goals). Further to this, on page 8 it states impact on attrition in graduate school.

• The manuscript is quite lengthy. While there are places where content is repeated and/or could be better synthesized, there are other areas where important information is missing. For example, on page 10 additional detail on how the authors selected participants from the potential pool of students, and ages of participants (if available) would also help readers determine the generalizability of findings. Ethical considerations are also dense and deserve some additional details. In places the discussion repeats information already reported in the findings.

• The limitations of the Appraisal Theory should be addressed, namely its reliance on individual cognitive appraisal, which is inherently subjective - should be included in the limitation section.

• Minor issue – there is some switching back and forth between current and past tense. Authors should do a careful review of the manuscript to ensure consistency in tenses.

• Page 8-9 identifies that half participants changed careers during graduate school while the other half changed careers after grad school. It is possible that career changes after graduate school were the result of other events or opportunities (and not something that took place in graduate school). This possibility should be addressed, especially since the researchers selected 10 participants from a larger group. Was this intentional to have students who had career changes after graduation? (this comment is similar to a previous one I provided).

• Additional detail is required on the recruitment and sampling technique. Specifically, description of the purposive sampling technique is needed. There are only 2 non-white participants and 2 Master’s students – was this intentional?

• Page 12 states “participants were provided pseudonyms based on color” – does this imply color of their skin? It is not clear what colors the authors are referring to. Or is it a case the pseudonyms are random colors selected to represent particpants?

• Was data from the draft interview protocol used for the 4 interviews included in the final study?

• Line 224-226 – it is not clear what is meant by “participants were guided that the event must be unexpected or unplanned” – does this mean the interviewer guided the participants i.e. tried to alter their perceptions of the event?

• Lines 268-269 – should this refer to career aspiration?

• Line 278 – suggestion writing “Research Question” out in full to be consistent with how this is presented further down in the manuscript.

• Line 530 the theme resiliency – I am not sure the illustrations provided supports resiliency.

• Line 659 The link between the study findings and career choice satisfaction among undergraduates needs to be established.

**Do you want your identity to be public for this peer review?** For information about this choice, including consent withdrawal, please see our Privacy Policy

Reviewer #1: **Yes: ** Billy Vinette

Reviewer #2: **Yes: ** Dr. Rose McCloskey RN PhD

---

## [Author Response · Author response to Decision Letter 1]

20 Jun 2025

Monitoring Editor: Patricia Morris

Thank you for the opportunity to review this paper. Overall I find the topic quite interesting, and the writing is sound. Both reviewers speak to the need for clarity in the theoretical framework and for contextualizing/clarifying the concept of chance events. Reviewer One questions whether the theoretical framework you have used is appropriate for the analysis. Reviewer Two suggests there are issues with the concepts of chance events and career aspirations/intentions/trajectories. This revision will require significant reworking to address the reviewers' concerns.

Thank you for your submission.

Review Comments Response To Reviewer

Reviewer 1

Lines 73–75: The authors state that nearly all participants recall an event that may have influenced their career choice. While this is certainly noteworthy, the assertion appears somewhat reductive. The degree to which such an event may have influenced career trajectories can vary significantly across individuals.

It would be important to highlight that it is not merely the presence or absence of such an event that matters, but also the context in which it occurred, the extent of its impact, and whether it triggered a cascade of subsequent chance events.

Furthermore, the nature of the chance event itself is not discussed. For instance, a chance event such as receiving unexpected funding might have a substantially different impact compared to meeting a particular person. This section would benefit from a more nuanced and in-depth discussion of these aspects. Yes, we agree that the impact of chance events on careers can vary. We have clarified this in the manuscript and added the previously published results of the survey from which the interview participants in this study were drawn. In the survey we collected information on types of chance events, the number experienced, the perceived valence (positive, negative, neutral), and the perceived level of impact on career intentions (low, medium, high).

We added additional information in the introduction about these results (now lines 80-94 and 185-187) to highlight the variety of ways in which chance events are experienced by graduate students and the gaps that led to this study. From this work, we documented that there were no patterns related to the type of chance event and impact level. For example, some participants experienced the same event with different perceptions of impact, or different events with similar perceptions of impact. These findings led us to focus on and investigate perceptions and reactions to chance events that were reported by participants as being impactful to their careers, without focusing on the nature of the event.

Therefore, we recruited participants from that survey sample, asked them to talk about chance events that had impacted them, and asked them to describe in detail one chance event that they perceived as being the most impactful on their career intentions. This was the chance event that was analyzed for this study. We have added a few more details about the interviews to clarify this (lines 218-244).

We added additional information about the limitations of methods and potential future work in lines 785-827.

Lines 79–81: Similarly, this passage does not offer insight into the broader context or the lasting impacts of the events described. The argument could be strengthened by elaborating on how contextual elements mediate or moderate the influence of these events. Yes, we agree that the context of participants is important - both as individuals with unique backgrounds and experiences and as individuals embedded in a graduate student community. We have added information to acknowledge this at lines 68-73 and 107-111.

However, we also clarified that this study was not designed or able to probe the impacts of these contextual factors. The intent of the analysis was to focus on individual emotional and cognitive processing and perceived impacts after the event happened. Lines 130-141 and 186-189.

The interviews for this study provided some information about context, asking them what was happening around the time of the event, how they reacted to and processed it, and its impact. However, because that was not the focus of this study, we did not collect enough information to consider participants' contexts within their programs, nor long-term effects. Nor did we have a large enough sample to try to generalize any of these impacts.

We also added additional information on participant context (lines 251-255) and this limitation to our study on lines 810-827.

Lines 81–83: While I agree with the authors on the importance of better understanding these processes, the discussion does not sufficiently emphasize the contextual factors in which emotional and cognitive responses occur. The focus seems to rest primarily on the individual's internal responses, with limited consideration of environmental factors (e.g., physical, economic, social) that may shape or constrain those responses.

We agree, but made a methodological and analytical choice to focus more narrowly on the individual processing and impact based on the findings from our previous study. We have added a note about this in the limitations / future directions (lines 810-827) to acknowledge that we did not have the data to fully explore the contextual influences you are asking about.

Lines 93–95: This section emphasizes cognitive processes, which is relevant. However, it would be valuable to incorporate a discussion of how individuals assess both internal resources (e.g., capacity to tolerate uncertainty) and external resources (e.g., availability of social support). A more holistic perspective could enhance the theoretical depth of the analysis. Certainly, our results indicate that students use internal resources (self-talk) and external resources (such as peers), both types of coping strategies. This information is in the discussion and we added more clear information about coping in the introduction (lines 113-119) to foreshadow these two elements of the findings.

Theoretical Framework: Lines 110–111: The authors highlight that the interpretation of events varies according to personal perceptions. While this point is valid, it may conflate individual perception with the broader context. An individual may indeed interpret an event in a particular way, but this perception does not negate the influence of the stable or shared context in which the event occurs. A clearer distinction between individual perception and contextual conditions would strengthen this argument. Yes. We do agree that student perceptions are shaped by their personal (e.g., health), social (e.g., relationships with peers/faculty/mentors/family), and institutional (culture or ability to access resources) contexts. We added information about this here (130-150) and other locations, as highlighted above.

Lines 117–118: The mention of “available resources” appears here for the first time, without any prior discussion in the introduction. Given the centrality of available resources in shaping individuals' interpretations of chance events, it would be important to revise the introduction to incorporate a more explicit discussion of their role and implications. Thank you for mentioning this. We expanded on internal and external resources in the intro (lines 113-119).

Lines 153–154: This statement appears to partially repeat the content presented in lines 98–102. To enhance clarity and avoid redundancy, I would suggest retaining only the formulation in lines 153–154, which is more concise and better integrated. After some consideration, we have decided to keep some information at the original lines 98-102 (now lines 120-124) but edited for clarity and to address the research questions. We believe it is important for the reader to have a preview of the study goals before the theoretical framework

Lines 159–160: The number of graduate students interviewed is mentioned in this section, yet this detail directly pertains to the research questions and design. As such, it would be more appropriate to present this information later in the methods section, rather than here. We changed this to “we interviewed life science graduate students…” at line 190. The term “life science graduate student" is defined in methods (now lines 222-225).

Methods: Overall: The methodology section is generally well-structured and clearly written. However, several key clarifications are necessary to enhance transparency and rigor.

COREQ Checklist: The manuscript would benefit from explicitly referencing and aligning with the COREQ (Consolidated Criteria for Reporting Qualitative Research) checklist. This would help ensure completeness and adherence to established standards for qualitative research reporting. Thank you for this. We have included strategies used to ensure transparency and reflexivity based on best practice noted by the COREQ checklist lines 307-310.

Line 169: The term “life science graduates” is introduced without a prior definition or justification. It would be important to clarify why this specific population was selected. What makes their experiences with chance events distinct compared to other graduate students? This choice raises questions regarding the transferability of findings to other academic fields.

Life science graduate students are defined in lines 222-225).

We added our reasoning for this choice of populations in lines 218-240. We did not select them because we thought they would have a unique response, but because of our familiarity with their context. We added at lines 785-868 that this is a small sample and that these findings are not generalizable.

Contextualization of Chance Events: The discussion of chance events lacks historical, socioeconomic, and sociocultural contextualization. Situating the participants within such broader contexts would provide readers with a more nuanced understanding of how these events are experienced and interpreted. We agree the context of participants is important. We added the contextual information we did collect in lines 284-287. This included participants' department and their experiences in the context of the COVID-19 pandemic.

We also acknowledge the limitations of not having a full analysis of student context and highlight this in the limitations and future work section (line 810-827).

Line 186: The manuscript states that “five individuals from this group” were selected, but the selection criteria and process are not described. Clarifying how participants were chosen would improve transparency and allow readers to better assess potential biases. Thank you for this. We clarified how we selected participants and when we closed recruitment (lines 233-244). Specifically, we grouped participants based on whether they changed their career intention since starting graduate school.

Lines 194–195: The authors state that “verbal consent was appropriate for this work.” This raises concerns, particularly as participants were initially recruited following a survey. It is unclear whether participants were provided with a full explanation of their rights and the scope of the qualitative interviews. There is a distinction between simply obtaining verbal consent (e.g., a “yes”) and securing informed verbal consent based on a detailed explanation of risks, benefits, and participant rights. This point requires further elaboration and ethical justification. We followed the IRB protocol which had a consent form common to the survey and interview. Students read the information and consented during that part of the study, and then verbal consent was a re-affirmation of the same information. We realize we did not make this clear and have clarified this in lines 210-217.

Line 199: Both master’s and PhD students were included in the study, yet these programs can differ greatly in duration, structure, and disciplinary culture. The rationale behind including such heterogeneity in participants’ academic backgrounds should be clarified. What are the implications of this choice for the interpretation of the results? Yes. This is important to clarify. While these degree types may be extremely different at other institutions, master’s and PhD programs at our institution are similar in structure, with the primary difference being the number of written chapters required. Thus, we did not distinguish between them for this study. This was added in lines 225-2328.

Lines 199–211: Much of the content in this section appears to describe findings or characteristics of the participants and may be more appropriately placed in the results section rather than the methods. Thank you, while we understand this rationale, we decided to keep this information at its location because it provides background information necessary to understand the recruitment for the interviews.

Lines 201–203: The sociodemographic profile of the participants appears quite homogeneous. This raises important questions about the diversity of experiences captured in the study. Were any deliberate efforts made to recruit participants outside the 25–34 age range? The contexts and chance events experienced by individuals in different life stages may vary significantly. We did attempt to recruit a diverse sample for the survey, but we have a fairly homogenous graduate student population at our institution. Most of the survey respondents who volunteered for interviews were within the 25–34 age range, which is reflective of the typical graduate student population at this institution. We acknowledge this as a limitation of the study and added suggestions for future work (lines 796-798), but the study was not meant to be generalizable across all graduate students.

Lines 215–218: It is unclear why the interview guide was not directly informed by the theoretical framework presented earlier in the manuscript. Instead, the authors indicate it was developed based on prior survey responses. A stronger justification is needed here, particularly regarding the relevance and limitations of this approach. Yes, we agree our wording was not clear here. The interview guide was guided by the theoretical framework, but the specific interview questions were framed around the chance events that the participants had mentioned in their survey responses. We have clarified this at lines 266-273.

Lines 220–223: It is noted that interview questions were developed using participants’ own survey responses. This raises ethical concerns: Did participants explicitly consent to this kind of data linkage and use? The overlap between the survey and interview phases may require additional ethical safeguards and greater transparency in the consent process. Yes. As mentioned above, the IRB approval was for both the survey and the interviews because of the linked nature of the studies.

Line 234: The manuscript refers to HF’s role in the research, but does not sufficiently describe HF’s experience with conducting qualitative research. This information is relevant for assessing the credibility of data collection and analysis and should be expanded. This information was added at lines 296-307.

Lines 244–248: I appreciate the authors’ transparency regarding the presence of a peer who conducted the interviews. However, this presence may have introduced social desirability bias, potentially influencing participants’ responses. In future studies, it might be advisable to involve an independent interviewer to minimize such bias. We understand this perception, but also wanted students to feel comfortable in their discussions. We definitely see pros and cons with each choice.

Use of Reflexivity Tools: There is no mention of the use of reflexive tools such as a research journal or analytic memos. Incorporating these tools is a widely recommended practice in qualitative research to support transparency and rigor, particularly in the interpretation of data. This was added at line 307-310.

Results: Overall: The results section presents compelling and relevant findings; however, it would benefit from further synthesis and conciseness. At 14 pages, it currently feels overly lengthy, which may hinder readability and reduce the clarity of the main insights. Thank you, we read and reduced verbiage throughout a

---

## [Decision Letter · Decision Letter 1]

13 Aug 2025

A Qualitative Study of Graduate Student Emotional and Cognitive Processing of Unexpected (Chance) Events

PONE-D-25-11301R1

Dear Dr. Ferguson,

We’re pleased to inform you that your manuscript has been judged scientifically suitable for publication and will be formally accepted for publication once it meets all outstanding technical requirements.

Kind regards,

Patricia Anne Morris

Academic Editor

PLOS ONE

Additional Editor Comments (optional):

Reviewers' comments:

Reviewer's Responses to Questions

**Comments to the Author**

Reviewer #1: All comments have been addressed

Reviewer #2: All comments have been addressed

2. Is the manuscript technically sound, and do the data support the conclusions?

Reviewer #1: Yes

Reviewer #2: Yes

3. Has the statistical analysis been performed appropriately and rigorously?

Reviewer #1: N/A

Reviewer #2: N/A

4. Have the authors made all data underlying the findings in their manuscript fully available?

Reviewer #1: Yes

Reviewer #2: Yes

5. Is the manuscript presented in an intelligible fashion and written in standard English?

Reviewer #1: Yes

Reviewer #2: Yes

Reviewer #1: I would like to thank the authors for addressing my comments and for providing thoughtful responses to my questions. Overall, I find that all of my concerns have been adequately addressed.

My only remaining suggestion pertains to the length of the manuscript, which has increased slightly with the addition of responses to the reviewers’ comments. I would encourage the authors to consider making the manuscript more concise, particularly by shortening or truncating some of the longer verbatim excerpts (e.g., […]), where appropriate.

Reviewer #2: The authors have addressed all of my concerns shared with my original review. The authors have done a commendable job with their revisions, particularly in the introduction and theory sections.

**Do you want your identity to be public for this peer review?** For information about this choice, including consent withdrawal, please see our Privacy Policy

Reviewer #1: **Yes: ** Billy Vinette, RN, Ph.D.

Reviewer #2: **Yes: ** Dr. Rose McCloskey

---

## [Editor Report · Acceptance letter]

PONE-D-25-11301R1

PLOS ONE

Dear Dr. Ferguson,

I'm pleased to inform you that your manuscript has been deemed suitable for publication in PLOS ONE. Congratulations! Your manuscript is now being handed over to our production team.

Kind regards,

on behalf of

Dr. Patricia Anne Morris

Academic Editor

PLOS ONE